# Pathways to mental health services across local health systems in sub-Saharan Africa: Findings from a systematic review

Samuel Adeyemi Williams[1]*, Mamadu Baldeh[1,2,3]*, Abdulai Jawo Bah[1], Frida Dennis[4], Dimbintsoa Rakotomalala Robinson[2,3], Yetunde C. Adeniyi[5]

1 College of Medicine and Allied Health Sciences, University of Sierra Leone, Freetown, Sierra Leone, 2 Medical Research Council Unit The Gambia at London School of Hygiene and Tropical Medicine, Serrekunda, The Gambia, 3 Clinical Research Department, London School of Hygiene & Tropical Medicine, London, United Kingdom, 4 Ministry of Health and Sanitation, Government of Sierra Leone, Freetown, Sierra Leone, 5 College of Medicine, University of Ibadan, Ibadan, Nigeria

☯ These authors contributed equally to this work.
* samadewill22@yahoo.co.uk (SAW); mbaldeh@gmail.com (MB)

## Abstract

Globally, over 280 million individuals suffer from mental disorders, and almost 85% in low-resource settings do not receive any therapy. In sub-Saharan Africa (SSA), many patients are forced to either live with untreated mental illness or seek care from traditional or religious leaders due to the high treatment cost. This literature review identifies pathways to access mental health services and proposed a collaborative model for care across SSA. We systematically searched five electronic databases (Embase and MEDLINE via OVID, CINAHL, PsycINFO, and Global Index Medicus) using the following search terms, 'pathways to care', 'mental disorders,' and 'sub-Saharan Africa' for primary studies reporting on pathways to care for mental disorders in SSA. There were no restrictions on the study's date. Overall, the electronic database search produced 3399 search results, of which we retrieved 194 articles for full-text screening and 29 studies included in the analysis. This study finds that traditional and faith-based healers play an integral role in the pathway to care; more than 70% used traditional and religious healers as the first point of care for mental health care. The median duration for the delay in seeking treatment in a health facility was six months. Patients who sought care from traditional and faith healers were found to have experienced the most prolonged delay without treatment. Age, gender, level of education, marital status, and geographical location were some of the factors associated with the pathway choice. Patients who sought care from traditional and faith healers as the first point of care were found to have experienced the most extended delay without treatment when they arrived at the hospital. The study proposes and recommends a new model for collaboration between biomedical, traditional and faith-based healers that focuses on education through training and adopting a new referral framework.

**Data availability statement:** All relevant data are within the paper and its Supporting files.

**Funding:** The author(s) received no specific funding for this work.

**Competing interests:** The authors have declared that no competing interests exist.

## Introduction

Mental health has been a low priority in most low and middle-income countries (LMIC), as most healthcare programs focus on infectious and non-communicable diseases [1]. Although there has been a growing awareness of the importance of mental health as a critical component of health in general, especially in child development, and probably due to the tremendous change in children's and adolescents' health and disease patterns [1,2], mental health is still left out of most health policy agendas across Sub-Saharan Africa (SSA) due to competing priorities with other healthcare demands, poverty, and conflicts. This is also reflected in the World Health Organization (WHO) Mental Health report 2022 [3], which shows considerable disparities in resource allocation (financial and human) for mental health in LMIC. The estimated average governmental budget for mental health across SSA is only around 2% of the total health budget [4]. Moreover, 60% of this expenditure is directed towards outdated approaches in psychiatric hospitals, with an out-of-pocket expenditure of over 40% [5].

The global burden of mental disorders varies mainly due to the heterogeneity in disease classification and methods used in measurement [6]. The WHO estimates a worldwide prevalence of 14% of the global burden of diseases attributable to mental disorders, with 75% of people affected in LMICs not receiving adequate treatment [7–9]. Based on Years-Lived-with-Disability (YLDs) and Disability-Adjusted Life-Years (DALYs) measurements of disease burden, Vigo *et al.,* 2016 [6] estimate that mental disorders account for 32.4% of YLDs and 13% of DALYs, making mental disorders the most burdensome disease in terms of YLDs. Although these estimates show the extent of the global burden of mental disorders over the years, the much-needed attention from stakeholders to place appropriate interest in prioritizing funding and treatment for persons with mental disorders is almost inexistent [10] and may therefore lead to neglect, stigma, and discrimination [11,12].

In LMICs, few skilled professionals are available to cater to the needs of mental disorders [5,13]. This lack of human resources is not limited to a particular country but is widespread throughout SSA. In Kenya, with a population of 50 million, there are only 45 psychiatrists, and only one is a trained child and adolescent mental health professional [14]. In Tanzania, there are 0.04 psychiatrists per 100,000 people and in Nigeria, there are 0.09 psychiatrists per 100,000 people [5]. In many parts of SSA, persons with mental disorders resort to seeking care from traditional or faith-based healers influenced by multiple factors [15–18]. The WHO estimates a ratio of 1:500 traditional & faith-based healers to the population compared to 1:40,000 doctors/population across SSA [18]. This disparity in available human resources, especially at the community level, influences health-seeking behaviour and commonly held traditional beliefs.

According to Ojagbemi *et al.*, 2021 [19], over two-thirds of patients suffering from mental disorders in SSA may concurrently seek traditional healers for mental health therapy, even after receiving successful hospital treatments. Many indigenous people are known to strongly embrace and promote traditional beliefs that inform health-seeking and healing practices, including beliefs and practices related

to sorcery and bewitchment [20,21]. On the other hand, scientists have recently recognized the significance of long-held traditions and religious-cultural norms as important to understanding the causes and treatments of mental disorders in sub-Saharan Africa [22–24].

An essential component of this review is to understand the patient's pathways to care through their varied help-seeking behaviours. It will help discern the pathway people take toward care, identify barriers and reasons for the delay, and thus inform policy and practice. Given the recent introduction of integrated primary healthcare services across SSA, which includes mental health services, it is important to review existing knowledge on the patterns of patients' help-seeking behaviour. A systematic literature review will help identify existing care paths across SSA and understand the influence on timely referral, provide helpful guidance for public mental health efforts [25]. It is, therefore, imperative to critically assess the roles different stakeholders play, whether formal or informal, at the time of the first consultation to determine if there is any significant contribution to delay in seeking formal healthcare services and assess the time it takes from experiencing the first symptoms or signs and accessing care. Elucidating on the different pathways to care will help understand the barriers that hinder access, including recursive pathways, and identify opportunities for enhanced collaboration with various stakeholders involved in mental healthcare service delivery.

This systematic mapping of pathways to mental healthcare service focused on the pathways to care based on the perceived cause of mental disorders in SSA and the duration of seeking care, and was guided by three research questions:

Research question 1: What are the existing pathways to care for individuals with mental health disorders in SSA?

Research question 2: What socio-demographic characteristics correlate with the pathways to care for people with mental health disorders in SSA?

Research question 3: What factors influence the duration of seeking treatment for mental health disorders in SSA?

In this study, we aim to systematically review existing literature on the pathways to care for patients with mental disorders across SSA to inform the design of a model pathway-to-care approach in SSA.

## Methods

### Search strategy

Following the guidelines outlined in the Preferred Reporting Items for Systematic Reviews and Meta-Analyses (PRISMA) checklist (S1 Table), we conducted a comprehensive systematic review to identify the existing pathways to healthcare for mental disorders. The protocol for this study is registered with Prospero, ID number CRD42023459738 and can be accessed at: https://www.crd.york.ac.uk/prospero/display_record.php?ID=CRD42023459738. Our search approach adhered to the Cochrane Collaboration guidelines. With the support of a librarian and mental health experts (Y.C.J. and S.A.W), we developed the search strategy for studies assessing mental health care pathways in SSA. Three independent reviewers (M.B., F.D. and S.A.W.) examined studies, specifically looking for direct pathways to care for mental disorders in SSA.

The primary search strategy involved an exhaustive search of academic databases using subject headings and keywords with MeSH terms to find relevant studies. We also hand-searched reference lists of all relevant articles and journals to identify any missed literature that could contribute to the research goal. For any new keyword or term identified, an additional search was done in the databases, and relevant papers were identified until no new article was found. We identified studies on 'pathways to care,' [26–28] 'mental disorders,' [18,29] and 'sub-Saharan Africa' using multiple keywords based on previous studies (S2 File).

We employed several targeted search strategies, including boolean operators, to ensure the robustness of the results from different databases (S3 Table). We searched the following databases: Embase, MEDLINE, CINAHL, Web of Science and Global Index Medicus.

## Inclusion and exclusion criteria

The criteria for inclusion and exclusion were established before the database searches began (S4 Table). Articles had to be (1) peer-reviewed, published, original research studies using qualitative, quantitative, or mixed methods design, (2) prospectively or retrospectively reported on perceived or measured barriers to access to mental healthcare services, and (3) treatment or help-seeking behaviours for mental disorders. Studies on related and sometimes overlapping concepts were also included, such as (4) standardised tools or specific methods to assess mental health services.

In this review, we define the pathway to care as a structured, multidisciplinary approach designed to facilitate coordinated decision-making and the organization of care for a specific patient population over a defined time frame. The aim is to improve the quality of care by improving risk-adjusted patient outcomes, ensuring patient safety, increasing patient satisfaction, and optimizing resource utilization across the continuum of care [30]. Furthermore, Rickwood *et al.* 2012 [31] define help-seeking pathway as an active adaptive process persons take to seek assistance in dealing with mental disorders, and this course is not random but guided by psychological and sociocultural factors. We defined 'mental disorder' or 'mental illness' as a clinically significant disruption in a person's behaviour, emotion control, or thought processes that reflects a breakdown in the biological, psychological, or developmental processes that underlie mental functioning [32–35]. Usually, these conditions are linked to severe discomfort or impairment in social, professional, or other crucial spheres of life. We covered mental healthcare services in SSA, including knowledge-based and non-knowledge-based pathways.

We excluded articles that were (i) researched from outside SSA, (ii) published in languages other than English or French, and (iii) Single-model consultation studies with either traditional/faith-based healers only or biomedical consultation only and were not related to the patient care journey. We did not restrict the study's date, and articles that were not available in full-text format were excluded (S5 Table).

## Heterogeneity, robustness and bias assessment

The recruiting strategies of the included studies varied by mental disorders and the first point of contact in the pathway continuum of care. While some studies discussed the pathway to care for specific mental disorders, others focused on the pathway to care for mental disorders as a single condition. Therefore, the studies were categorized based on pathways reported for analysis.

We excluded studies from the analysis that the preliminary checks involve participants who have sought care from traditional healers before, but the pathways are not described to ensure uniformity. Also, studies that did not focus specifically on mental health conditions as an inclusion criterion were excluded from the analysis. In the case of unclear or missing data, the authors were contacted to obtain further clarification or additional information. Only studies with distinct proportions of mental health conditions involving various care pathways were included in the analysis.

## Quality assessment and data extraction

We used an adapted assessment tool to evaluate the methodology and quality of reporting quality of the included studies (S6 Table), which integrated elements from the Cochrane Collaboration's critical appraisal tool for qualitative studies [36], the Strengthening the Reporting of Observational Studies in Epidemiology (STROBE) guidelines [37], and the Consolidated Criteria for Reporting Qualitative Research (COREQ) [38]. The adapted tool focuses on capturing the documented eligibility criteria, descriptions of methodologies and findings, sampling strategies, internal validity, and the generalizability of the results. The methodological quality of three types of research was evaluated: qualitative research, quantitative descriptive studies, and mixed methods studies. Two authors independently conducted this quality assessment. When discrepancies arose in the quality ratings, a third author was consulted to facilitate resolution through discussion. Percentage scores were utilized to categorize the quality of evidence: (i) 50% indicates low-quality evidence, (ii) 51–75% indicates average-quality evidence, and (iii) 76–100% indicates high-quality evidence.

Three research team members independently screened our records using titles and abstracts. We retrieved the full texts of documents that required further review based on our inclusion and exclusion criteria. We reviewed all the papers included and discussed how to reach a consensus in cases of disagreements. In cases where the two reviewers could not decide on an included paper, a third reviewer was consulted to reach a consensus. The degree of concordance among screeners' findings while reviewing abstracts and complete articles was assessed by computing Cohen's kappa statistics. The interpretation of kappa statistics is as follows: values <0.1 signify no agreement, 0.10–0.20 suggest none to slight agreement, 0.21–0.40 denote fair agreement, 0.41–0.60 indicate moderate agreement, 0.61–0.80 signify substantial agreement, and 0.81–1.00 represent almost perfect agreement.

All references were managed using the Endnote® software. References and PDFs were organized in a specialized folder with comments and annotations. Papers that met the inclusion criteria were examined thoroughly for content familiarization and contributions to the question were extracted in a review. Each article involved in the review was critically appraised. The appraisal started with trying to answer the six questions (where, how, when, what, who, and why) developed by Wooliams et al., 2011 [39]. For each paper, we extracted (i) publication details: title, author, year, institution, and (ii) descriptive details: Study context, location, mental disorder, pathway to care, and duration to seeking care, which is presented in a descriptive format. Recurred themes from qualitative and mixed-methods studies with noted. Data extraction was done using a Microsoft Excel spreadsheet. Both reviewers rechecked extracted data to address any disagreements.

### Data analysis

We employed convergent and explanatory mixed-methods models to synthesize and integrate descriptive analysis and thematic findings. Most included studies used the WHO Encounter Form from the WHO Pathways to Care initiative. We review the findings from each study to explore the interlinked pathways to care.

Jain et al. (2012) [40] explored three pathways to care for mental disorders: traditional healers, specialists, and physicians. In this review, we explored these three pathways and counted the number of consultations sought, the time delay until accessing specialist care, and the roles of stakeholders. We categorized and tallied the study settings, participants, sample size, and data collection methods and presented quantitative data as descriptions and proportions.

For collecting and reporting qualitative findings, several iterative interactions were conducted to discuss the scope of the review per standard guidelines. To synthesize the results, we employed the "Synthesis Without Meta-analysis" (SWiM) guideline [41].

To adequately capture context-specific and sensitive aspects during a patient's journey within the care pathway, an iterative and inductive narrative synthesis approach was used to integrate evidence from the various methodological study designs, including quantitative and qualitative studies addressing pathways to mental health. We used the adjusted version of the Levels and Filters Model by Goldberg & Huxley (1996) [42] to categorize our findings. The pathway progression from community to specialized mental health services and the occurrences highlight the transition between various mental health care levels. The levels correspond to the mental disorders discussed, those who seek help from primary care services, patients who get referred to specialized mental healthcare services, and patients who ultimately receive specialized care. These levels are complementary, and analysis was done to reflect the SSA context.

## Results

### Included studies

Table 1 summarizes all papers meeting the inclusion criteria, the study setting and participants, the sample size, study design, data collection method, and quality assessment.

**Prisma flow diagram.** Fig 1 summarizes the article selection process using the PRISMA flow diagram. Overall, the electronic database search produced 3399 search results: 1294 from Embase, 1144 from MEDLINE, 852 from Web

**Table 1. Summary characteristics of included studies.**

| Study | Country | Setting | Sample (sub-group)/ [N] | Study type (data collection) | Aim |
|---|---|---|---|---|---|
| Abdulmalik et al., 2012 [25] | Nigeria | Tertiary psychiatric facility | Children and adolescents [N = 242] | Quantitative survey (questionnaire) | To promote awareness about mental disorders and encouraging early presentation for treatment |
| Bakere et al., 2013 [46] | Nigeria | Neuropsychiatric facility | Children and adolescents [N = 393] | Quantitative survey (questionnaire) | Assess first points of contact during help seeking and eventual sources of referral |
| Kamau et al., 2017 [14] | Kenya | Tertiary mental health clinic | Children and adolescents [N = 166] | Quantitative survey (questionnaire) | Determined the psychiatric morbidity and socio-demographic profile of patients who eventually present for care |
| Abiodun et al. 1995 [43] | Nigeria | General hospital | Adolescents and Adults (>16) [N = 238] | Quantitative survey (WHO encounter form) | Examined routes to modern mental health care in Nigeria for a wider range of patients. |
| Adeosun et al., 2013 [44] | Nigeria | Neuropsychiatric Hospital | Adults [N = 138] | Quantitative survey (SSI questionnaire) | Assess the pathways to mental health care among patients with schizophrenia at their first contact with mental health services |
| Patel et al., 1997 [63] | Zimbabwe | Primary health care clinics | Adolescents and Adults (15–70) [N = 53] | Quantitative survey (POC questionnaire) | Described the pathways to primary care for patients with common mental disorders |
| Temmingh et al., 2008 [58] | South Africa | General hospital (Psychiatry department) | Adults [N = 71] | Quantitative survey (WHO encounter form) | Investigated pathways to care in an ethnically diverse group of inpatients with psychotic disorders. |
| Odinka et al., 2014 [55] | Nigeria | Neuropsychiatric Hospital | Adults [N = 367] | Quantitative survey (WHO encounter form) | Assessed the influence of sociocultural factors on help-seeking behaviours among patients with schizophrenia and their association with duration of untreated psychosis (DUP) |
| Nonye et al., 2009 [54] | Nigeria | Neuropsychiatric hospital | Adolescents and Adults (15–75) [N = 397] | Quantitative survey (Questionnaire) | Determined the health-seeking behaviour of mentally ill patients |
| Lasebikan et al., 2012 [53] | Nigeria | General hospital (Psychiatric unit) | Adolescents and Adults (14–58) [N = 652] | Quantitative survey (Questionnaire) | Determined the relationship between social network and pathway to service utilization among psychotic patients |
| Kauye et al., 2014 [65] | Malawi | Psychiatric units | Adolescents and Adults (13–75) [N = 128] | Quantitative survey (WHO encounter form) | Understand prior care-seeking and treatment of new patients seen at mental health services in a developing country |
| Girma et al., 2011 [67] | Ethiopia | Specialized Hospital | Adults [N = 384] | Quantitative survey (WHO encounter form) | Investigated patterns of treatment seeking behavior and associated factors for mental illness. |
| Appiah-Poku et al., 2003 [64] | Ghana | Teaching hospital (Psychiatric unit) | Adults N = 302 | Quantitative survey | Identified previous help sought by patients presenting to the services for an initial assessment. |
| Aghukwa et al., 2012 [45] | Nigeria | Tertiary hospital | Adults N = 219 | Quantitative survey (WHO encounter form) | Examined treatment seeking of psychiatric patients |
| Bekele et al., 2008 [62] | Ethiopia | Specialized Mental Hospital | Children and Adults (2–85) N = 1044 | Quantitative survey (WHO encounter form) | Described the routes taken by patients to reach psychiatric care, evaluate the time delay before seeking psychiatric care, and investigate the relationship between delay on the pathway to care and sociodemographic and clinical factors. |

*(Continued)*

**Table 1.** (Continued)

| Study | Country | Setting | Sample (sub-group)/ [N] | Study type (data collection) | Aim |
|---|---|---|---|---|---|
| Burns et al., 2011 [61] | South Africa | Psychiatric referral hospital | Adolescents and Adults (17–45) N=54 | Quantitative survey | Investigated the relationship between spiritual/traditional attributions of illness causation and/or a history of previous consultation with traditional healers prior to hospital admission. |
| Ibrahim et al., 2016 [66] | Ghana | Psychiatric hospital | Adults N=107 | Quantitative survey (WHO encounter form) | Understand the pathways that people with mental disorders traversed for psychiatric services, and the factors that influence such pathways to mental health care. |
| Lund et al., 2010 [60] | South Africa | Psychiatric hospitals | Adults N=152 | Quantitative survey (SSI) | Examined service utilization patterns and pathways to specialist mental health services among individuals with schizophrenia spectrum disorders |
| Mkizie et al., 2004 [57] | South Africa | Mental health institution | Children and Adults (10–59) N=15 | Qualitative (SSI) | Determined pathways of care the clients with mental illness take, which ultimately lead to the mental health institution, the effects of socio-cultural and economic factor on the pathways to mental health care and the satisfaction with different service providers consulted. |
| Modiba et al., 2001 [56] | South Africa | Community clinic | Children and Adults (8–81) N=68 | Mixed methods (Survey & SSI) | Investigated community mental health service needs of mental health service users and that of their families |
| Jack-Ide et al., 2013 [52] | Nigeria | Neuropsychiatric hospital (outpatient clinic) | Adults N=50 | Qualitative (IDI) | Explored the pathways to mental health service of families and persons with mental health issues before they arrive at mental health care services. |
| Gureje et al., 1995 [49] | Nigeria | Tertiary psychiatric service | Adults N=159 | Qualitative (SSI) | Identification of sources of delay in the receipt of care and suggest possible improvements |
| Gureje et al., 2006 [50] | Nigeria | Community (household) | Adults N=4984 | Quantitative (Diagnostic Interview) | To highlight pattern and determinants of mental health service use in the community |
| Erinosho et al., 1977 [48] | Nigeria | Mental Hospital; and Community Village Programme. | Adults N=208 | Quantitative survey | Distinguishes referral processes from referral sources. |
| Galvin et al., 2023 [22] | South Africa | Psychiatric facilities | Adults N=309 | Qualitative (SSI) | Assessed the perceptions and experiences of mental illness and treatment among patients with mental illness |
| Tomita et al., 2015 [59] | South Africa | Tertiary psychiatric hospital | Adults N=57 | Quantitative survey (WHO encounter forms) | Examined first-contact patterns and pathways to psychiatric care among individuals with severe mental illness |
| Odinka et al., 2014 [68] | Nigeria | Mental health services. | Adults N=360 | Quantitative survey (WHO encounter forms) | To assess the association between the positive and negative symptoms of schizophrenia, help-seeking and DUP |
| Bella-Awusah et al., 2020 [47] | Nigeria | Child psychiatry and paediatric neurology clinics | Children N=114 | Quantitative (WHO encounter forms) | To identify and compare the pathways to care for children and adolescents |
| Ikwuka et al., 2016 [51] | Nigeria | Community | Adults N=706 | Quantitative survey | Explored the pathways to care for mental illness preferred by a non-clinical sample of the population |

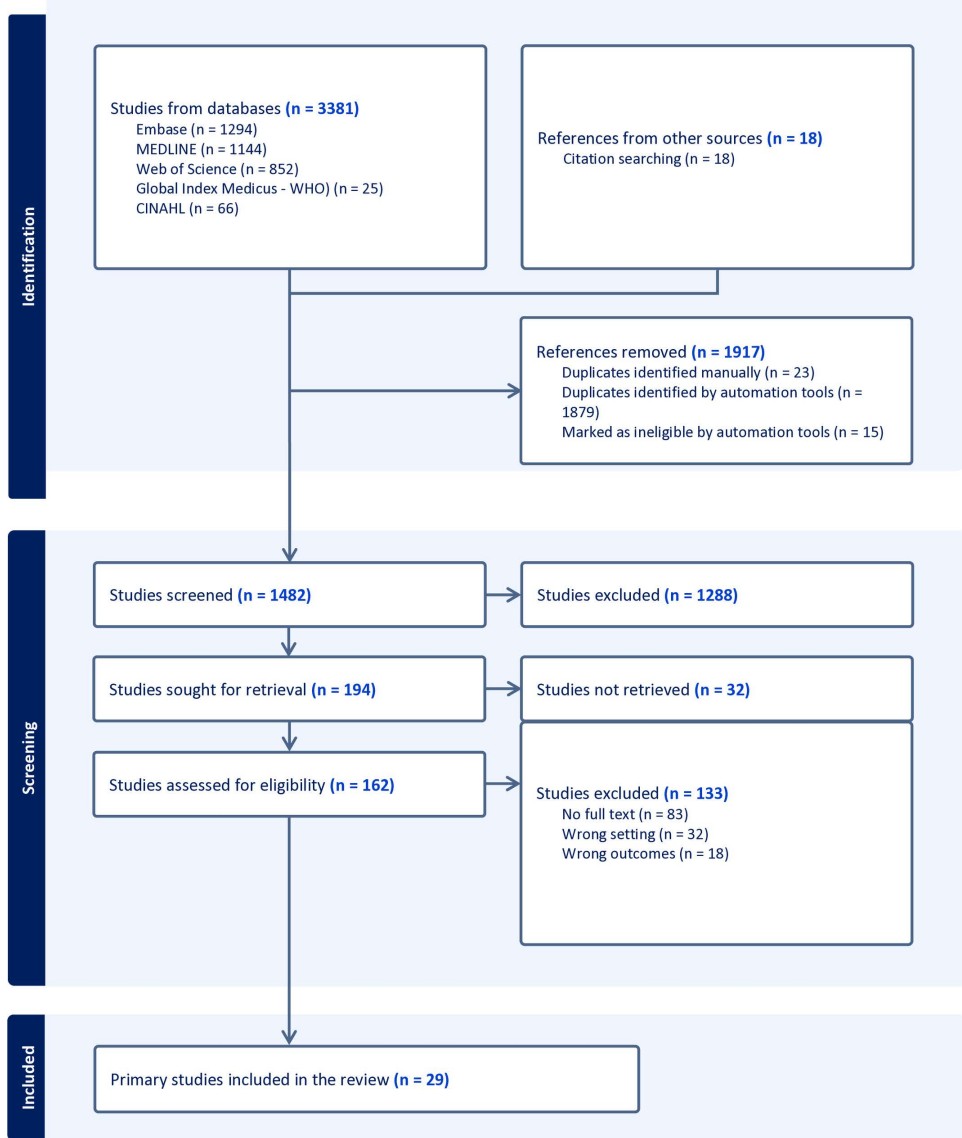

**Fig 1. Prisma flow diagram.**

of Science, 66 and 25 from CINAHL & WHO-Global Index Medicus, and 18 citation handsearching. Results were then exported to Endnote, where manual screening was done for additional duplicates. After the deduplication process, 1482 papers were screened through the title and abstract. We retrieved 194 articles for full-text screening, of which twenty-nine studies (24 quantitative, 4 qualitative and 1 mixed-methods) met the inclusion criteria, four exclusively focusing on children and adolescents and twenty-five with varying populations, primarily adults. All included studies were in English.

**Chronology of included publications per year, January 1977 – September 2023.** Fig 2 presents the articles that were included based on the inclusion criteria. All included studies were peer-reviewed articles. We identified one study from 1977, two studies from 1995, and one study from 1997, which is the earliest published literature in this review. Seven

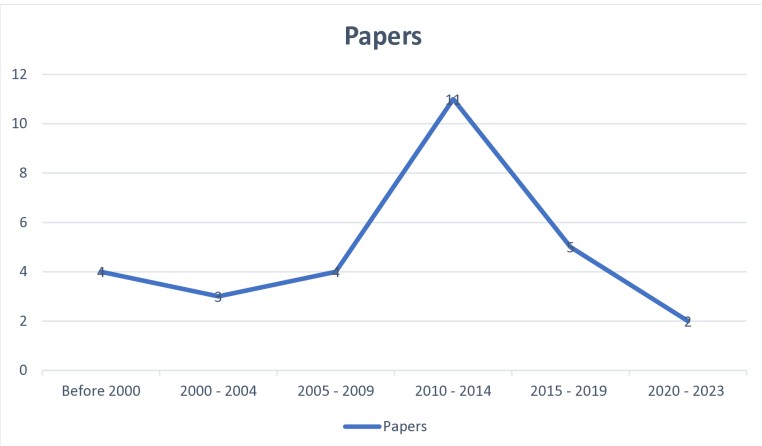

**Fig 2. Chronology of included publications.**

studies were included between 2000 and 2009, while sixteen were from the following decade, spanning 2010–2019. We identified and included two additional studies during the COVID-19 period spanning 2020–2023.

**Geo-spatial locations of included studies.** This review draws from three of the four regions in SSA (East, South, and West Africa) and includes studies from seven countries. About 58% (17/29) of the articles are from West Africa, specifically Ghana and Nigeria, with Nigeria having the bulk (fifteen) of the articles in the review. East Africa (Ethiopia, Kenya, and Malawi) has five studies, while Southern Africa (South Africa and Zimbabwe) has eight studies. South Africa contributed eight studies, the second highest number after Nigeria. We did not find any study from Central Africa that was eligible. More than 70% of the studies were located in two countries, Nigeria [25,43–55] and South Africa [22,56–61]. The most extensive individual study, however, is from Nigeria [62], with a sample size of 4984 participants, representing about 39.9% of the total sample size (total participant number) of all the included studies (N = 12,491).

**Research context and approach.** Fig 3 presents the study methods employed by each article. Fifteen studies utilized various interviews (three semi-structured, eleven structured interviews, and one clinician-administered interview), nine used the WHO Encounter form, four used a modified/ adapted WHO Encounter form, and one used the Pathways to

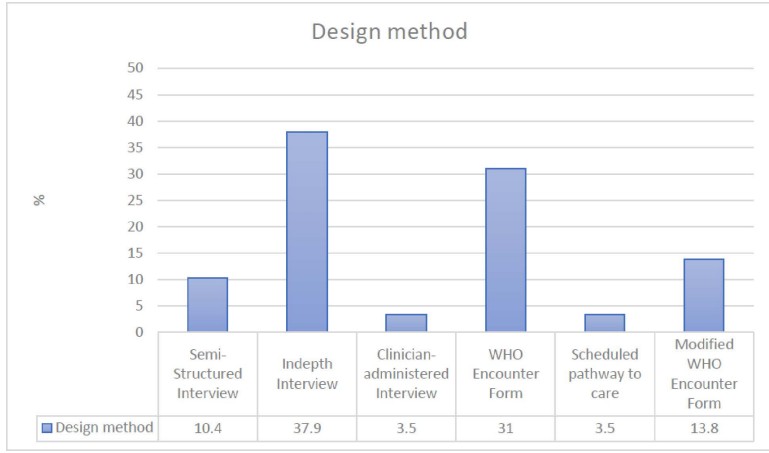

**Fig 3. Reported study methods.**

**Table 2. Details of care pathway consulted per study.**

| Study | Traditional healers (%) | Religious leaders/ healers (%) | Traditional/ Religious (undifferentiated) (%) | General practitioners (GPs) (%) | Direct tertiary (%) | Police (%) | Others (%) |
|---|---|---|---|---|---|---|---|
| Abdulmalik et al., 2012 [25] | – | – | 36.4 | 3.3 | 60.2 | – | – |
| Bakare et al., 2013 [46] | 6.9 | 22.4 | – | 20.6 | 47.6 | – | Patent stores, special schools: 2.6 |
| Kamau et al., 2017 [14] | – | 6 | – | 31.9 | 52.4 | – | Pharmacy: 0.6 Juvenile justice system: 0.6 School counselor: 3.6 Change environment: 4.2 Relatives: 0.6 School assessment centres: 1.8 |
| Abiodun et al., 1995 [43] | 26.5 | 13.4 | – | 55.9 | – | – | Patent medicine dealers: 4.2 |
| Adeosun et al., 2013 [44] | 11.6 | 42.8 | 14.5 | 13.8 | 17.4 | – | – |
| Patel et al., 1997 [63] | 24.5 | 10 | – | 21.9 | – | – | Primary care clinic: 42.7 |
| Temmingh et al., 2008 [58] | – | – | 5.6 | 39.4 | 29.6 | 25.4 | – |
| Odinka et al., 2014 [55] | 14.7 | 61.1 | – | 14.7 | 6.4 | – | Drug store: 3.1 |
| Nonye et al., 2009 [54] | 13.6 | 34.5 | – | 15.9 | 32 | – | Patent medicine vendor: 4 |
| Lasebikan et al., 2012 [53] | – | – | 78.9 | 9.3% | 4.5 | – | – |
| Kauye et al., 2014 [65] | – | – | 26.1 | 46.9 | – | 2.7 | Psych nurse: 24.3 |
| Girma et al., 2011 [67] | 20.1 | 30.2 | – | – | 35.2 | – | Biomed Institute: 13.3 |
| Appiah-Poku et al., 2003 [64] | 5.9 | 14.2 | – | 52.8 | 2 | – | Family doctor: 4.7 |
| Aghukwa et al., 2012 [45] | 28 | 69 | – | 24 | – | – | Other health professionals: 3 |
| Bekele et al., 2008 [62] | 4.5 | 30.9 | – | 21.5 | 41 | – | Psych nurses: 2 |
| Burns et al., 2011 [61] | 38.5 | – | – | 16 | – | 44 | – |
| Ibrahim et al., 2016 [66] | – | – | 23.3 | 21.5 | 52.3 | – | Community health nurse: 2.9 |
| Lund et al., 2010 [60] | – | – | – | 73 | 29 | 51 | – |
| Mkizie et al., 2004 [57] | 20 | 20 | – | 33.3 | – | – | District Hospital: 20 Primary Healthcare clinic: 6.7 |
| Modiba et al., 2001 [56] | 30 | – | | 70 | – | – | Friends: 7 |
| Jack-Ide et al., 2013 [52] | 20 | 48 | – | 20 | 12 | – | – |
| Gureje et al., 1995 [49] | 19 | 13 | – | 47 | 20 | – | – |
| Gureje et al., 2006 [50] | – | – | 3.7 | 28.2 | 29.1 | – | 32.8 |

*(Continued)*

**Table 2.** (Continued)

| Study | Traditional healers (%) | Religious leaders/ healers (%) | Traditional/ Religious (undifferentiated) (%) | General practitioners (GPs) (%) | Direct tertiary (%) | Police (%) | Others (%) |
|---|---|---|---|---|---|---|---|
| Erinosho et al., 1977 [48] | 0.5 | – | – | 4.3 | 3.4 | – | Family: 89.4 |
| Galvin et al., 2023 [22] | 28.16 | 15.04 | 4.7 | 53 | – | – | – |
| Tomita et al., 2015 [59] | – | – | 11.5 | 3.9 | 73.1 | 1.9 | Clinic: 3.9 Social Worker: 1.9 Private Psychiatrist: 3.9 |
| Odinka et al., 2014 [68] | 14.7 | 61.1 | – | 17.8 | 6.4 | – | – |
| Bella-Awusah et al., 2020 [47] | – | 54.39 | – | 31.58 | 14.04 | – | – |
| Ikwuka et al., 2016 [51] | 33.2 | 57.8 | – | 90.8 | – | – | – |

Care Schedule. Four studies focused exclusively on the pathways to care for children and adolescents: three in Nigeria [25,46,47] and one in Kenya [14]. The other studies primarily focused on adult and caregiver populations, while some included children as young as two years old [62]. Burns *et al.* 2011 [61] only asked if participants had sought care from traditional healers before; details of other pathways are not described.

## Pathways to care

Table 2 presents various routes that patients and caregivers take to access mental health services for mental disorders. The various routes of consultation can be broadly classified into formal and informal pathways.

This review identified formal pathways as health facilities: primary health care clinics, general and regional hospitals, and tertiary mental health hospitals; and medical personnel: general practitioners (GPs), nurses, pharmacists, drug stores, and psychiatrists. Informal routes include traditional healers, religious healers, and unregistered medical vendors. Specialized schools and the juvenile justice system were also used as a pathway to care for children and adolescents [14,47].

**Pathways to care for children and adolescent.** This review finds that the most frequently used care pathway by caregivers for children and adolescents was the direct route to tertiary hospitals, followed by general health services and practitioners.

The estimate for caregivers who seek mental health services directly from tertiary care ranges from 47.6% to 60.2% from three studies on children and adolescents [14,25,46]. This is followed by those seeking care in general medical services, with an estimated 20.6% to 37.9% [14,46]. Only 3.3% of caregivers sought care from general practitioners in Nigeria [25].

A considerable number of people, however, consulted traditional or religious healers as their first choice of care. Results from these four studies on children revealed an average of 29.3% to 47.1% of caregivers consulted traditional or religious healers as their primary source of care, except in the study by Kamau *et al.* 2017 [14], where only 6% of patients or caregivers contacted religious leaders. Their study nevertheless identified other varied and important routes taken by patients/ caregivers in need of help. These routes include consulting school counsellors, accessing the juvenile justice system, and seeking pharmacy help.

**Pathways to care for adults.** Twenty-five studies were analyzed for pathways to care in the adult population. This review shows that general medical services, tertiary hospitals, and traditional/ religious healers were the main pathways to care used by patients with mental health disorders.

While participant in some studies alluded to the preference of biomedical health services for treatment modalities [43,49,56–58,62–66], most participant showed a preference for traditional and religious healers especially studies conducted in Nigeria over the last fifteen years, with the earliest published in 2009. In contrast, the other two that preferred biomedical services were conducted in 1995.

In the studies that found a preference for traditional & faith-based healers, 78.9% of the study population consulted traditional healers as the first source of care [53]. Conversely, in Temmingh & Oosthuizen, 2008 [58] study, only 5.6% of participants consulted them.

In another study by Burns et al. (2011) [61] in Kwa-Zulu Natal, South Africa, 38.5% of patients admitted at the tertiary hospital sought care from traditional healers as the first point of care, with only 35% seeking care from biomedical services (19% from psychiatrists).

This review, however, notes that many people used several other help-seeking pathways for mental health service, including the police (2.9% to 25.4%) [56,58,65], nurses (2% to 21.1%) [62,65], patent medicine vendors/ pharmacies (3.1% to 4.2%) and other community health workers [14,43,46,54].

## Socio-demographic correlations of pathways to care

**Studies on children and adolescents.** Children accompanied to the psychiatric facility were found to be two and a half times (OR=2.5, 95% CI 2.35–2.6) more likely to have had a mental illness for less than six months compared to children who were accompanied by a single parent [25].

In the study by Kamau (2017) [14], females and children without disruptive disorders were associated with pathway choice. Female children were more likely to be taken through the biomedical pathway by their caregivers than males (p = 0.027), and this was the same for children without disruptive behaviours compared to those with disruptive behaviours.

**Studies on adults.** Of the twenty-five studies focused on adult and mixed-age groups, eleven found at least one factor to be significantly associated with the choice of pathway. These factors include age, sex, marital status, education, geographic location (rural or urban), perceived ideas about the cause of the illness, family, employment, and aggressive behaviours.

In the study by Odinka et al. 2014 [55], participants who were below the age of 40 years were significantly more likely to use faith-based healers as their first option for a pathway to care when compared to those above 40 years who preferred traditional healers as their first option [55]. Another study found participants who were between the ages of 31 and 40 years to be more than ten times less likely to seek treatment when compared to those below the age of twenty (OR-10.7, 95% CI 1.99–56.99) [67].

Patients who have spent more than six years in formal school were more likely to use the biomedical pathway as the first option than those with fewer years in formal school [55]. This was similar to the findings by Nonye et al. 2009 [54], who established that a higher level of education is associated with specialist service usage as their first pathway to care. Patients who lacked formal education experienced a significant delay from starting symptoms to reaching the formal healthcare service for mental health treatment [62].

In terms of location, patients living in rural communities were found to be more likely to use traditional medicine when compared to people from urban communities [68]; this was also supported by Nonye et al. 2009, where he made a significant observation that urban community dwellers are more likely to use specialist mental health service compared to those in rural communities. Attribution of the perceived causation of mental disorders based on location also played a role in the pathway to care. Correct rationalization of the cause of mental disorders was associated with consultation with specialist medical services [54]. Furthermore, traditional and spiritual attribution of causation and previous consultation with traditional healers were associated significantly with a long delay of untreated psychosis [61].

Regarding employment status, self-employed patients have almost twice the odds of seeking mental health care in general medical health facilities and more than four times higher odds with public servants than unemployed patients [66]. People who were unemployed/ jobless were also found to experience significantly longer times (delays) in accessing

mental health care from the onset of symptoms to the first point in the care pathway and then reaching a formal mental health service [62]. Other important correlates that were significant in determining access to formal mental health services and early treatment seeking were male gender [54], a family member with psychiatric illness, marital status, presence of somatic symptoms [67], and those in full contact with family almost every day [53].

## Duration/delay in seeking treatment

The distribution measurement for the delay in seeking care was highly skewed in most reported studies; therefore, the median was used. For children and adolescents, the median delay from the onset of symptoms to accessing care at a formal health service was between 4.5 and 54 months [inter quartile range (IQR) = 22.6 months] [14]. In another study, 64.5% of patients at a psychiatric hospital in Nigeria were found to have been with a mental illness for more than six months before presenting to the hospital [25].

Evidence from studies on the adult population indicates that the overall median delay from symptom onset to presentation at the hospital was between 4.5 and 9.5 months [58,62,64], with an IQR between 35–37 months. Aghukwa *et al.* 2012 reported a mean of 54 months from disease onset to formal mental health consultation. When the different pathways that were sought were factored in, a patient who first consulted traditional or faith-based healers were found to have experienced the most extended delay when they arrived at the hospital compared to those who consulted formal health services directly, and this association was significant [43,44,49,55,62,64,65].

## Patients/caregivers' perception of the cause of mental illness and sources of referral

The perception of causes of mental illness can be broadly classified into spiritual, environmental, and medical causes, with some participants admitting that they do not know. Spiritual belief emerged as one of the main reasons patients consulted with traditional and faith-based healers. In a study conducted in Nigeria, 71% to 85% of patients consulted with and preferred traditional or faith-based healers because they believed in the supernatural origin of the mental disorder [44,45], and this was reported by only 3.2% of patients in Zimbabwe [63]. On average, 29.1% to 69.4% of caregivers believed that mental disorders are due to spiritual causes [45,54,61,67].

The evidence also suggests that 22.3% to 41% of caregivers believed in the biomedical causation of mental disorders [45,54,61], between 19.6% to 48.6% acknowledge that they have no idea about the causes of mental illness [54,67], and 16.9% associated them with environmental factors [45].

## Role of stakeholders and decision-making on provider and service

This domain addresses the factors and belief systems that influence the first treatment decision to seek care. These factors include traditional ideas on the cause of mental disorders, religious attitudes regarding treatment, stigma and discrimination, and inadequate awareness of mental health services. The decision on the choice of the first point of contact in the care pathway is almost entirely influenced by made by family members, including parents, spouse, brother, and close friends in the community. Police services have also been recorded as being involved in the transfer of patients seeking formal medical services [57].

Tradition or faith-based healers play a significant part in the mental healthcare pathway in SSA. In the study by Jack-Ide *et al.*, 2013 [52], 20% of the participants believed that supernatural forces or hostile spells were the source of mental health problems, which could only be cured by pleasing the gods and required traditionalists and herbalists to carry out specific sacrifices and cleansing.

*"'We went for traditional treatment, we have taken her to so many places herbalist or whatever. We have done so many things because of this illness…at a point the healer asked us to bring various denomination of money from N5 to N1,000 each, we brought them and these monies were burnt as sacrifice in the name of treating her" (Caregiver) [52].*

Early mental health diagnosis, according to service users, reduced the length of an episode, decreased the direct and indirect expenses of treating the illness, and significantly lessened the family's long-term social damage. Individuals had their general practitioners refer them to a mental health facility after they had previously sought help from friends or family [52]. Patients who sought treatment from a primary healthcare clinic and those who had more household family members were more likely to have sought treatment from traditional healers. This suggests that patients are more prone to turn to traditional healers rather than biological treatment when they live in more crowded and impoverished environments [22]

"I was advised to bring her here by a doctor at…a medical doctor referred us to this place and that was how we got here. Since then she has improved greatly, the changes are amazing" *(Caregiver)* [52]

## Discussions

This study reviewed the evidence in the literature on the pathways to mental healthcare services in SSA. It also explored the socio-demographic correlations that influence help-seeking behaviour and suggested a collaborative model for seeking care. The findings in this review mainly reflect medical pluralism as reported for different mental health challenges in SSA. The fact that biomedical specialist care is scarcely available for consultation may be related to the limited human resources for formal healthcare services in these countries. This highlights the need for increased training and decentralized mental healthcare service delivery.

Because patients and caregivers use a combination of biomedical and traditional or faith-based pathways to access mental healthcare services, this strongly calls for a collaborative model for mental healthcare service delivery in the SSA. The synthesized studies showed that many patients who accessed formal mental healthcare services had consulted with traditional or faith-based healers as the first point of care. This finding is similar to that reported by the WHO and other previous studies [8,69].

Most studies in this review reported an overall preference for biomedical mental health services as a preferred treatment option; however, many patients would have consulted with the informal sector before the biomedical option, and it also varied with education, age, and gender.

Six of the eight articles that showed an overall preference for traditional or faith-based healers were from Nigeria, with one each from Ethiopia and South Africa, respectively. This should, however, be considered specific to cultural factors or causal attributions that might influence this preference in the pathway to care and in SSA. Recursive pathways exist where patients move between formal and informal services [44,57,62,65,67], consistent with findings from other LMICs [70–72].

### Delay in access to care

As alluded to earlier, one of the most important aspects in considering help-seeking behaviour must be how the choice of first care provider affects or leads to delays in accessing appropriate treatment. Delays in treatment result in increased morbidity and mortality, including significant harmful health effects to both society and the individual (e.g., substance abuse, psychiatric commodities, and life-altering self-treatments).

This review found that patients who sought care from traditional and faith-based healers as the first point of care were found to have experienced the most prolonged delay in accessing evidence-based mental health services at tertiary hospitals. Complicated correlation patterns indicate multiple causative factors for delay, such as educational attainment, income status, availability of healthcare services, and cultural barriers. The lack of knowledge regarding mental health is a significant concern that requires adequate assessment and action plan. Strategies that reduce delays in accessing mental healthcare services and early interventions in patients with mental illnesses will decrease the morbidity associated with mental disorders [73,74].

The delay in care highlighted in this review should inform strategies on how formal biomedical services and stakeholders should engage with traditional or faith-based healers and other informal health providers in SSA to improve the

pathways to care for persons with mental disorders. Even though the aim is to shorten the delay by getting patients into formal health services quickly, it is difficult to achieve this in these settings where resources are limited, and structures or services to aid this process are not readily available in the communities [71]. Inadvertently, the mental health needs of patients, caregivers, and communities are unmet by the existing health policies and services [75,76].

## Collaborative model

This review proposes that the pathway to care for mental health disorders would benefit from similar models used in other public health programs, such as the TB and HIV programs that have successfully implemented collaborative care models with traditional or faith-based healers in some SSA countries [77–80]. Other collaborative models based on the views of traditional and biomedical practitioners to address mental health treatment gaps have been proposed in rural Kenya [81], Liberia [82], and Zanzibar, Tanzania [83]. These models align with the recommendations by Patel, 2011 [84], which suggested a collaborative model as the ideal approach for global mental health service delivery. Our review has evaluated several of the context-specific collaborative models to draw insights and inform a comprehensive collaborative mental healthcare model for SSA.

Stakeholders in formal care pathways have raised concerns about the safety and efficacy of interventions employed by traditional or faith-based healers and reports of human rights abuses against people with mental disorders [18,85]. It should also be noted that cases of abuse are not peculiar to traditional or faith-based healers, as there have been harrowing reports of human rights abuses even in tertiary mental health facilities [86,87]. On the other hand, traditional healers have also expressed concerns about working with healthcare workers. These stem from the fear of reproducing their treatment techniques and concoctions in scientific labs [88,89]. Another concern is that biomedical practice is considered to be superior to traditional healing techniques, and therefore, the notion that the latter should not be recognized [90].

Therefore, implementing a proposed collaborative model cannot be thoroughly carried out if these concerns are not adequately addressed. A collaborative model aims to foster a working relationship between the two care systems and implement a clear referral pathway. It will also serve as a platform where the reasons for 'unsafe' interventions are investigated and researched, with proper solutions and behavioural change implemented. This model aims to put the needs and welfare of the patient at the centre, where there is a shared responsibility between the providers they choose to consult, which is facilitated by mutual respect and understanding. Formal mental health services can also use this opportunity to learn from traditional or faith-based healers' psychosocial approach, given that the main goal is to harness treatment modalities that can promote the health of patients with mental health problems. The model does not attempt to interfere with the duties of both traditional and biomedical services but rather train traditional or faith-based healers on evidenced-based psychosocial interventions and open referral pathways between the two services, Fig 4.

The WHO Mental Health Global Action Programme intervention guide (MhGAP-IG) intervention guide was designed to be used by non-specialists in health facilities at primary care levels. It recognizes traditional and faith healers as non-specialists and a potential resource to reduce the treatment gap in low- and middle-income countries [4,91]. However, MhGAP must be adapted to suit their local context, resources, priorities, and other mental disorders not initially included.

A collaborative model herein will enhance the easy and quick identification of mental health cases at the community level by community healthcare workers and traditional and faith-based healers. The new model will also enhance and build upon weak referral pathways by collaborating with these community members to identify persons with mental health problems, offer appropriate evidence-based psychosocial interventions, and refer complex cases directly to community primary care clinics or tertiary hospitals.

## Strengths and limitations

The strength of this review is the potential to shape standards for healthcare service delivery involving mental health care and enhance the optimization of access to mental healthcare services. They may interest a wide range of stakeholders,

**Fig 4. Pathways to care and a collaborative model.**

including researchers and policymakers, all aiming to comprehend the fundamental concepts behind access to mental healthcare services. Increased awareness and knowledge could enable decision-makers to contribute to developing strategies that promote a smoother policy and implementation approach. The mixed methods model applied to the findings in this review offers a comprehensive picture of pathways to mental healthcare services and to understand the nuances associated with complex and under-resourced healthcare settings. The findings may apply to other health contexts because of multiple mental health disorders, participant groups who are prone to have heightened symptoms and settings. Therefore, our findings emphasize the necessity of mental health screening in community settings through a collaborative referral model to enable referral and encourage favourable mental health outcomes. Another strength of this review is the reproducibility of the search strategy and protocol registered with it, which can be accessed in PROSPERO.

Our review has limitations, excluding grey literature and those published in languages other than English, which may have led to selection bias and reduced the pool of relevant studies. The studies included in this review presented results

from cross-sectional help-seeking behaviour using standardised questionnaires to assess the pathways to care. Therefore, we acknowledge that our analysis may represent a scenario where only a fraction of the actual path and access to mental healthcare was reported.

Another critical limitation stems from the fact that almost all the studies in this review were conducted at a tertiary mental health facility, with a few conducted in other formal health services such as clinics. Many people with mental health disorders may have never accessed formal mental health services. Therefore, their help-seeking behaviour is limited to only traditional or faith-based consultation and could not be fully reflected in this review as a pathway to care.

## Conclusion

This review proposes and recommends a new model for collaboration between biomedical and traditional or faith-based healers that focuses on education through training and adopting a new referral framework. The proposed collaborative model's goal is to harness the practices of traditional or faith-based healers that promote health to benefit the mentally ill. It is also important to note that the focus on improved access to mental health care in SSA should focus on other innovative strategies. One such strategy, in addition to collaboration, should also focus on fully integrating mental healthcare into general and primary healthcare settings, with appropriate training of healthcare workers at the community level.

## Supporting information

**S1 Table. Prisma checklist.**
(PDF)

**S2 File. Search strategy legend.**
(PDF)

**S3 Table. Search strategy and output.**
(PDF)

**S4 Table. Selection criteria.**
(PDF)

**S5 Table. Characteristics of excluded studies.**
(PDF)

**S6 Table. Study quality assessment and appraisal.**
(PDF)

## Author contributions

**Conceptualization:** Samuel Adeyemi Williams, Mamadu Baldeh.

**Data curation:** Samuel Adeyemi Williams, Frida Dennis.

**Formal analysis:** Samuel Adeyemi Williams, Mamadu Baldeh, Frida Dennis.

**Methodology:** Mamadu Baldeh, Abdulai Jawo Bah.

**Resources:** Samuel Adeyemi Williams.

**Validation:** Samuel Adeyemi Williams.

**Visualization:** Samuel Adeyemi Williams, Mamadu Baldeh.

**Writing – original draft:** Samuel Adeyemi Williams, Mamadu Baldeh, Dimbintsoa Rakotomalala Robinson.

**Writing – review & editing:** Samuel Adeyemi Williams, Mamadu Baldeh, Abdulai Jawo Bah, Frida Dennis, Dimbintsoa Rakotomalala Robinson, Yetunde C. Adeniyi.

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
