## [Decision Letter · Decision Letter 0]

22 Jan 2025

Dear Dr. Baldeh,

Thank you for submitting your manuscript to PLOS ONE. After careful consideration, we feel that it has merit but does not fully meet PLOS ONE’s publication criteria as it currently stands. Therefore, we invite you to submit a revised version of the manuscript that addresses the points raised during the review process.

We look forward to receiving your revised manuscript.

Kind regards,

Ryan G Wagner, MSc(Med), MBBCh, PhD

Academic Editor

PLOS ONE

3. We note that Figure 3 in your submission contain [map/satellite] images which may be copyrighted. All PLOS content is published under the Creative Commons Attribution License (CC BY 4.0), which means that the manuscript, images, and Supporting Information files will be freely available online, and any third party is permitted to access, download, copy, distribute, and use these materials in any way, even commercially, with proper attribution. For these reasons, we cannot publish previously copyrighted maps or satellite images created using proprietary data, such as Google software (Google Maps, Street View, and Earth). For more information, see our copyright guidelines: http://journals.plos.org/plosone/s/licenses-and-copyright.

1. You may seek permission from the original copyright holder of Figure 3 to publish the content specifically under the CC BY 4.0 license. 

4. As required by our policy on Data Availability, please ensure your manuscript or supplementary information includes the following:

Additional Editor Comments:

Please address each comment raised by the three reviewers. Additionally, I raise the following considerations:

1.) Please ensure that you are referencing the most relevant, up-to-date literature. For example, in the most recently referenced article is from 2020 (even though the authors refer to the 2022 WHO Mental Health Atlas...) and new figures (particularly for the GBD study have been published). 

2.) As noted by the reviewers, carefully review the manuscript to remove grammatical errors, including punctuation, misspelled words and incomplete sentences. 

Reviewers' comments:

Reviewer's Responses to Questions

**Comments to the Author**

1. Is the manuscript technically sound, and do the data support the conclusions?

Reviewer #1: Yes

Reviewer #2: Yes

Reviewer #3: Partly

2. Has the statistical analysis been performed appropriately and rigorously?

Reviewer #1: No

Reviewer #2: Yes

Reviewer #3: N/A

3. Have the authors made all data underlying the findings in their manuscript fully available?

Reviewer #1: No

Reviewer #2: Yes

Reviewer #3: Yes

4. Is the manuscript presented in an intelligible fashion and written in standard English?

Reviewer #1: Yes

Reviewer #2: Yes

Reviewer #3: Yes

Reviewer #1: This paper is a valuable contribution to the field of mental health promotion, addressing an often-overlooked area in sub-Saharan Africa. The methodology is generally sound, but there are areas needing clarification, such as the inclusion/exclusion criteria, heterogeneity in the results, and publication bias considerations. Additionally, more emphasis on key findings in the abstract would strengthen the paper. The study is missing information on the contribution of each author in assessing the quality of selected papers and whether authors of published papers were contacted for further information. This would be important for transparency and assessing the quality of each selected paper. There are no concerns regarding dual publication or ethical issues. However, attention should be given to refining the presentation of results and ensuring consistent use of abbreviations and correction of typographical errors throughout the manuscript.

Reviewer #2: This is a potentially very interesting and significant study. However, there are many spelling and grammar errors. Just in the Abstract alone there are many unfinished sentences and problematic phrasings (ie. “Overall, twenty-nine We” and “six and fifty four months”). These need to be addressed before the manuscript can be seriously considered.

Line 355: “children and adolescents” missing S at the end

No need to use MH as mental health acronym. Just say mental health

INTRODUCTION: Need a paragraph about traditional African belief systems and why traditional healing is so common/popular in Africa, particularly for mental illness. Need to explain how difficult to treat ailments (thus mental illness) are perceived by many as “spiritual afflictions” and thus treatment requires the intervention of healers.

See: Galvin, M., Chiwaye, L., & Moolla, A. (2023). Perceptions of causes and treatment of mental illness among traditional health practitioners in Johannesburg, South Africa. South African Journal of Psychology, 53(3), 403-415.

Line 463 - citation needed

INTRODUCTION AND DISCUSSION: I think you need to be careful about the refrain you keep repeating that “the articles in the review showed an overall preference 468 for biomedical health services as a preferred treatment option” - many of the articles have found the opposite to be true. There needs to be more nuance in the suggestion that people prefer biomedical care particularly when the first line of care is usually traditional healers.

Reviewer #3: This manuscript contributes to understanding mental health pathways in sub-Saharan Africa and proposes a framework for collaboration between allopathic mental health services and alternative pathways. A few areas of clarification might improve the manuscript

1.A large number of empirical studies on health seeking behavior for mental health problems have been excluded from the review without a clear rationale. Here are links to a few examples of primary studies and reviews from where primary studies can be extracted https://www.thelancet.com/journals/lanpsy/article/PIIS2215-0366(15)00515-5/abstract
https://pubmed.ncbi.nlm.nih.gov/23595533/ . The authors may consider reviewing the search terms to include primary studies conducted outside of tertiary facilities.

2. The diagnostic category and severity of mental disorder greatly influences the pathway to care but this has not been defined or described in the methods, results or discussion. This is also important in refining the definition of pathways to care for MH because not all cases require tertiary mental health care and in fact, there is extensive literature on the effectiveness of task sharing for most prevalent disorders such as depression and anxiety. Additionally, most studies in the review were conducted in tertiary facilities which significantly reduces the generalizability of the findings to a very small proportion of people with mental health problems.

3. While the proposed model is promising, it lacks details on operationalization, funding, and scalability in resource-limited settings. Consider reviewing evidence from studies that have explored similar versions of the proposed model such as this study from Kenya https://link.springer.com/article/10.1186/s13002-015-0075-6

Minor comments

-Consider reviewing the manuscript for conciseness and completeness of information. Below are a few examples:

• The keywords do not match the overall theme of the article and might mislead readers searching for articles related to child and adolescent mental health or to task sharing interventions like the mhGAP

Abstract

• Review for completeness and clarity. Some sentences are incomplete (e.g line 32, the sentence Overall, twenty-nine ‘’ is left hanging) and others are unclear (e.g ‘The median duration for the delay in seeking treatment in a health facility was six and fifty-four months’) is it 6 months or 54 months?

Background

• Review for accuracy of information. For example line 77-78 states that “In low and medium-income countries including Africa, few skilled professionals are available’ . Africa is not a country.

• Consider using updated references, for example there is a more recent study from Kenya that found ~ 120 registered psychiatrists https://bmchealthservres.biomedcentral.com/articles/10.1186/s12913-023-09481-w

• Consider making the background concise, to highlight the gap that the review seeks to address

**Do you want your identity to be public for this peer review?** For information about this choice, including consent withdrawal, please see our Privacy Policy

Reviewer #1: **Yes: ** Emmanuel Biracyaza

Reviewer #2: No

Reviewer #3: No

---

## [Author Response · Author response to Decision Letter 1]

20 Feb 2025

General:

This paper is generally well-written and provides a much-needed systematic review of an overlooked

Mental health promotion is one of the SDGs' goals. The report is of good quality, but I have provided some recommendations for improvement.

• We thank the reviewers for their insightful feedback, which will lead to a better publication. We are grateful that they recognise the importance of this study. We provide a line-by-line response to the feedback below in blue, alongside a revised manuscript.

Abstract

1. You should start with a stronger statement that emphasises the global and sub-Saharan African (SSA) public health concerns related to mental disorders. It is important to highlight the research gap that led to your interest in conducting this study in SSA before introducing your study’s purpose.

• Thank you. We have added further detail highlighting the research gap in a global and regional context. Line 26-30: “Globally, over 280 million individuals suffer from mental health disorders, and almost 85% do not receive any therapy in low-resource settings. Mental illness significantly affect patients, families, and their communities. In sub-Saharan Africa (SSA), many patients are forced to either live with untreated mental illnesses or seek care from traditional or religious leaders due to the high treatment cost”.

2. Clarify the total number of articles your literature search yielded. After applying the

inclusion/exclusion criteria, how many studies were retained? Mention the databases used for

selecting eligible studies (e.g., Embase, MEDLINE, CINAHL, PsycINFO, and Global Index

Medicus, from line 151). Also, specify the time frame for the selected studies

• Thank you, this is a helpful suggestion. We have now edited and added the following sentence to line33-36: “We systematically searched five electronic multiple databases (Embase and MEDLINE via OVID, CINAHL, PsycINFO, and Global Index Medicus) using the following search terms, 'Pathways to care', 'Mental health,' and 'sub-Saharan Africa’ for studies.”

• “We did not restrict the study's date, and articles that were not available in full-text format were excluded.”

3. Lines 34-37 need stronger results—elaborate 3-5 key findings (e.g., factors) to capture and facilitate readers' attention.

• Thank you, this is a helpful suggestion. We have now edited and added the following sentence to the result section. “This study finds that traditional and faith-based healers play an integral role in the pathway to care; more than 70% used traditional and religious healers as the first point of care for mental healthcare. The median duration for the delay in seeking treatment in a health facility was six and fifty-four months. Patients who sought care from traditional and faith healers were found to have experienced the most prolonged delay without treatment. Age, gender, level of education, marital status, and geographical location were some of the factors associated with the pathway choice. Patients who sought care from traditional and faith healers as the first point of care were found to have experienced the most extended delay without treatment when they arrived at the hospital. The study proposes and recommends a new model for collaboration between biomedical, traditional and faith-based healers that focuses on education through training and adopting a new referral framework.”

Background

4. The background section is well-written.

• Thank you

Methods

5. Provide more details on the inclusion and exclusion criteria in lines 165-167.

• Thank you. We recognise the importance of clearly describing the inclusion and exclusion criteria. We have made changes and added the following sentences to this section: “The criteria for inclusion and exclusion were established before the database searches began. Articles had to be (1) peer-reviewed, published, original research studies using qualitative, quantitative, or mixed methods design, (2) prospectively or retrospectively reported on perceived or measured barriers to access to mental health care, and (3) treatment or help-seeking behaviours for mental health problems to be included. Studies on related and sometimes overlapping concepts were also included, such as (4) standardised tools or specific methods to assess pathways to care.

For this review, we define pathways to care for mental health disorders as any care points used before care is sought from a mental health professional and the factors that influence the decision-making process. Every type of mental healthcare pathway was covered, including knowledge-based and non-knowledge-based. The methods by which patients engaged with their health-seeking behaviour were not excluded.”

• “. We did not restrict the study's date, and articles that were not available in full-text format were excluded”

6. Did you rely solely on academic databases for information, or did you contact the studies' authors for unpublished data? Figure 1 suggests that additional sources such as WHO, programs,

theses and registers were used, but this is not satisfactorily stated in the methods section.

• Apologies. This is an important critique. We did not clearly describe this in the method section. As explained above in the inclusion criteria, we did not include grey literature in the analysis. However, we search multiple databases and reference searches for published studies. The WHO-Global Index Medicus was one of the databases searched, which is indicated in the PRISMA flow diagram as “WHO”. We have now revised the PRISMA flow diagram and made the corrections. The full description of the PRISMA flow diagram is in the result section.

7. Of the 29 studies, how many were qualitative, mixed methods, or only quantitative?

• Thank you, this is a very helpful suggestion. We have now reviewed and added the total study methods to the description of the “prisma flow diagram”: “We retrieved 197 articles for full-text screening, of which twenty-nine studies (24 quantitative, 4 qualitative and 1 mixed-methods) met the inclusion criteria.”

Results

8. In Table 1, change the title to “Summary characteristics of included studies” and add a column for study design. Include more details on methods used (e.g., specify the type of interviews, such as FGDs or KIIs). Consider adding a column for the study population (optional) and another column for the quality assessment of each study (weak, moderate, strong). Also, the key results for each study will be presented.

• Thank you. We have now incorporated this helpful suggestion, edited the title as suggested, and added columns of study population and quality assessment to Table 1. Table 2 contains the key findings from each included study.

9. Discuss the potential risk of publication bias and related biases. Include more details on

heterogeneity in your results, as it will help assess the reliability of the overall summary. Where heterogeneity is high, focus on high-quality studies to generate inferences.

• Thank you, this is a helpful suggestion. We have added a sub-section sentence to the methods section:

“Heterogeneity, Robustness and Bias assessment

The recruiting strategies of the included studies varied. Some studies only included individuals with varied mental health conditions and multiple pathways to care as a recruiting criterion. This is likely to skew the analysis. Therefore, the studies were categorised based on the conditions and care pathways reported for analysis.

Studies with a high risk of bias were excluded to assess the robustness of the results. We excluded studies from the analysis that the preliminary checks involve participants who have sought care from traditional healers before, but the pathways are not described to ensure uniformity. Also, studies that did not focus specifically on a mental health condition as an inclusion criterion were excluded from the analysis.

The analysis did not include studies with missing findings. Only studies with distinct proportions of mental health conditions involving various care pathways were included in the analysis.

10. Since you used a mixed methods systematic review, I am missing the link between qualitative and quantitative methods. Secondly, which type of mixed methods systematic review did you conduct? Please provide a table summarising the literature synthesis's themes and subthemes. Provide quotes synthesised in the results section.

• Thank you. We recognise that the original manuscript did not clearly describe this, so we have now clearly described this in the “Data analysis” section: “We employed both convergent and explanatory mixed-methods models to synthesise and integrate descriptive analysis and thematic findings. Most included studies used the WHO Encounter Form from the WHO Pathways to Care initiative. We review the findings from each study's semi-structured open-ended interviews to explore the interlinked pathways to care adopted before reaching the hospital.

Jain et al. (2012) explored three pathways to care: traditional healers, specialists, and physicians. This review explored these three pathways, counted the number of consultations sought, the time delay until accessing THC and roles of stakeholders.

“For collecting and reporting qualitative findings, several iterative interactions were conducted to discuss the selection of manuscripts and the scope of the review per standard guidelines. To synthesise the results, we employed a nine-item checklist called the "Synthesis Without Meta-analysis" (SWiM) guideline (Campbell et al., 2020).”

We have also revised the “result” section and added the following sub-section with quotes:

• “Role of stakeholders and decision-making on provider and service

This domain addresses the factors and belief systems that influence the first treatment decision to seek care. These factors include traditional ideas on the cause of mental health difficulties, religious attitudes regarding treatment, stigma and discrimination, and inadequate awareness of mental health services. The decision on the choice of a pathway or provider is made by family members, including their parents, spouse, and brother, and inputs are from the community members. Police services have also been recorded as being involved in the transfer of patients seeking formal medical services (Mkize et al., 2004)

Tradition or faith-based healers play a significant part in Africa's mental health care pathway. In the study by Jack-Ide et al., 2013, 20% of the participants believed that supernatural forces or hostile spells were the source of mental health problems, which could only be cured by pleasing the gods and required traditionalists and herbalists to carry out specific sacrifices and cleansing.

“We went for traditional treatment, we have taken her to so many places herbalist or whatever. We have done so many things because of this illness…at a point the healer asked us to bring various denomination of money from N5 to N1,000 each, we brought them and these monies were burnt as sacrifice in the name of treating her” (Caregiver) (Jack-Ide et al., 2013)

Early mental health diagnosis, according to service users, reduced the length of an episode, decreased the direct and indirect expenses of treating the illness, and significantly lessened the family's long-term social damage. Individuals had their general practitioners refer them to a mental health facility after they had previously sought help from friends or family (Jack-Ide et al., 2013). Patients who sought treatment from a primary healthcare clinic and those who had more household family members were more likely to have sought treatment from traditional healers. This suggests that patients are more prone to turn to traditional healers rather than biological treatment when they live in more crowded and impoverished environments (Galvin et al., 2023)

“I was advised to bring her here by a doctor at…a medical doctor referred us to this place and that was how we got here. Since then she has improved greatly, the changes are amazing” (Caregiver) (Jack-Ide et al., 2013)”

Discussion

11. Expand on the strengths of the study (e.g. firstly, the use of mixed methods should be a strength; Secondly, the study protocol will be published in PROSPERO, and a theoretical lens will inform the robust thematic analyses. Discuss the weaknesses, such as excluding studies from grey literature and those in other languages, which may have led to selection bias. The limited inclusion of studies published in languages like German, Portuguese, and Dutch could have reduced the pool of relevant studies. Address the generalizability of your findings, either in the strengths or limitations.

• Thank you. We have now edited the “Discussion’ sections to incorporate this helpful suggestion. We have added the following sentences to the “strengths and limitations”: “The mixed methods model applied to the findings in this review offers a comprehensive picture of paths to mental health care and to understand the nuances associated with a complex and under-resourced healthcare setting. These findings may apply to other health contexts because of multiple mental health disorders, participant groups who are prone to have heightened symptoms and settings. Therefore, to enable referral and encourage favourable mental health outcomes, our findings emphasise the necessity of mental health screening in community settings through a collaborative referral model. Furthermore, thr review protocol was registered in PROSPERO which can be access online for detailed search strategy.

Our review have limitations including the exclusion of studies from grey literature and those published in languages other than english, which may have led to selection bias and reduced the pool of relevant studies”

Minor comments

13. Be consistent with abbreviations for 'mental health' and 'sub-Saharan Africa,' ensuring their use throughout the paper.

• Thank you, we have now revised and made this change throughout the document

14. Proofread the study to correct typos and grammatical errors (e.g., line 241 should read “2023” instead of “20223”).

• Thank you, we have now revised and made this change throughout the document

15. Most of the included studies are observational, primarily cross-sectional. Did you have criteria to include other study designs, such as experimental or longitudinal studies, or quasi-experimental designs?

Also, correct the typo in line 32 by deleting the words “Overall, twenty-nine.”

• Thank you for these comments. While we focused on primary studies with varied study designs, most studies that met our inclusion were observational, cross-sectional studies.

• Thank you, we have now revised and made this change throughout the document.

Additional Editor Comments:

1.) Please ensure that you are referencing the most relevant, up-to-date literature. For example, in the most recently referenced article is from 2020 (even though the authors refer to the 2022 WHO Mental Health Atlas...) and new figures (particularly for the GBD study have been published).

• Thank you, we have now updated this reference list.

2.) As noted by the reviewers, carefully review the manuscript to remove grammatical errors, including punctuation, misspelt words and incomplete sentences.

• Thank you, we have now revised and made this change throughout the document

Reviewer #1

This paper is a valuable contribution to the field of mental health promotion, addressing an often-overlooked area in sub-Saharan Africa. The methodology is generally sound, but there are areas needing clarification, such as the inclusion/exclusion criteria, heterogeneity in the results, and publication bias considerations. Additionally, more emphasis on key findings in the abstract would strengthen the paper. The study is missing information on the contribution of each author in assessing the quality of selected papers and whether authors of published papers were contacted for further information. This would be important for transparency and assessing the quality of each selected paper. There are no concerns regarding dual publication or ethical issues. However, attention should be given to refining the presentation of results and ensuring consistent use of abbreviations and correction of typogr

---

## [Decision Letter · Decision Letter 1]

31 Mar 2025

Dear Dr. Baldeh,

We look forward to receiving your revised manuscript.

Kind regards,

Ryan G Wagner, MSc(Med), MBBCh, PhD

Academic Editor

PLOS ONE

Reviewers' comments:

Reviewer's Responses to Questions

**Comments to the Author**

Reviewer #1: All comments have been addressed

Reviewer #3: (No Response)

2. Is the manuscript technically sound, and do the data support the conclusions?

Reviewer #1: Yes

Reviewer #3: Partly

3. Has the statistical analysis been performed appropriately and rigorously?

Reviewer #1: Yes

Reviewer #3: N/A

4. Have the authors made all data underlying the findings in their manuscript fully available?

Reviewer #1: Yes

Reviewer #3: Yes

5. Is the manuscript presented in an intelligible fashion and written in standard English?

Reviewer #1: (No Response)

Reviewer #3: Yes

Reviewer #1: Dear Authors,

I would like to express my sincere appreciation for your dedicated efforts in revising this manuscript. The study presents compelling results that hold substantial value for the scientific community. The field of mental health is particularly crucial in today's context, where mental health concerns and their associated challenges continue to pose significant obstacles. This piece of work makes a meaningful contribution by addressing these pressing issues and advancing our understanding in this vital area.

Bests

Reviewer #3: I thank the authors for considering my comments and providing thoughtful responses.

1. However, the authors have not taken all reviewers concerns into consideration about reviewing the manuscript in its entirety and revising multiple typographical errors and completing sentences which are crucial to understanding the key points discussed in the manuscript. For example, in my original review, I recommended reviewing literature that has proposed collaborative models and I provided some references. While the authors say that they have looked at these references and included them in the discussion and introduction section, this has been done as incomplete sentences, therefore I am unable to assess whether the authors did indeed review these literature. For example, in the discussion, line 460-462 reads ""Although several theoretical models of pathways to care that predict the decision-making process and investigate factors have been proposed in several studies

(Musyimi et al., 2016; Herman et al., 2018: Solera-Deuchar et al., 2020)."". Although this phrase includes one of the references I suggested, it is an incomplete sentence and I am therefore unable to access the authors' argument in that phrase.

There are other errors throughout the manuscript.

2. One unique contribution of this paper is the collaborative model which the authors propose. However, as stated in my original review, it is unclear how this model can be operationalized or scaled up. In its present state, the study's suggested models may not be practical. For example, the study proposes training traditional healers to offer psychosocial support and mentions the mhGAP as a potential training tool. The mhGAP is a clinical protocol, designed to build capacity among healthcare practitioners who are already licensed by a recognized clinical governing body to practice in a given jurisdiction. Although the word "lay healthcare workers" is often used , it does not refer to unlicensed practitioners. Therefore the authors might reconsider reviewing this section.

**Do you want your identity to be public for this peer review?** For information about this choice, including consent withdrawal, please see our Privacy Policy

Reviewer #1: **Yes: ** EMMANUEL BIRACYAZA

Reviewer #3: No

---

## [Author Response · Author response to Decision Letter 2]

31 Mar 2025

A point-by-point response to Reviewers

We appreciate the reviewers’ thoughtful comments that will undoubtedly enhance the quality of our publication. Their acknowledgement of the significance of this research is heartening. Our detailed reactions to each of their comments are presented below, accompanied by an updated version of our manuscript. Some of the comments have been broken down into section to provide detailed response.

Reviewer #1

Dear Authors,

I would like to express my sincere appreciation for your dedicated efforts in revising this manuscript. The study presents compelling results that hold substantial value for the scientific community. The field of mental health is particularly crucial in today's context, where mental health concerns and their associated challenges continue to pose significant obstacles. This piece of work makes a meaningful contribution by addressing these pressing issues and advancing our understanding in this vital area.

Bests

• We thank you for the feedback and grateful that you recognised the importance of this study.

Reviewer #3:

I thank the authors for considering my comments and providing thoughtful responses.

• Thank you.

However, the authors have not taken all reviewers concerns into consideration about reviewing the manuscript in its entirety and revising multiple typographical errors and completing sentences which are crucial to understanding the key points discussed in the manuscript.

• We have reviewed the manuscript in its entirety and made detailed changes and responses to all reviewers' comments. Minor typographic errors may have been missed due to editing. We have corrected the manuscript for all typographical mistypes and incomplete sentences.

For example, in my original review, I recommended reviewing literature that has proposed collaborative models and I provided some references. While the authors say that they have looked at these references and included them in the discussion and introduction section, this has been done as incomplete sentences, therefore I am unable to assess whether the authors did indeed review these literature. For example, in the discussion, line 460-462 reads ""Although several theoretical models of pathways to care that predict the decision-making process and investigate factors have been proposed in several studies

(Musyimi et al., 2016; Herman et al., 2018: Solera-Deuchar et al., 2020)."". Although this phrase includes one of the references I suggested, it is an incomplete sentence and I am therefore unable to access the authors' argument in that phrase.

• Musyimi et al., 2016; Herman et al., 2018; Solera-Deuchar et al., 2020, discussed context-specific collaborative models in the countries of these studies. We reviewed them in the context of developing the collaborative model for SSA and also cited them.

• We have revised and rephrased this sentence to inform the collaborative model section: “Line 512-518: Other collaborative models based on the views of traditional and biomedical practitioners to address mental health treatment gaps have been proposed in rural Kenya (Musyimi et al., 2016), Liberia (Herman et al., 2018), and Zanzibar, Tanzania (Solera-Deuchar et al., 2020). These models align with the recommendations by Patel, 2011, suggested a collaborative model as the ideal approach for global mental health service delivery. Our review has evaluated several of the context-specific collaborative models to draw insights and inform a comprehensive collaborative mental healthcare model for SSA”

• We included other suggested references, which were considered to support the introduction and or the discussion sections.

There are other errors throughout the manuscript.

• Thank you. We have revised the manuscript and made changes where required.

One unique contribution of this paper is the collaborative model which the authors propose. However, as stated in my original review, it is unclear how this model can be operationalized or scaled up. In its present state, the study's suggested models may not be practical.

• While our proposed collaborative model is informed by reviewing primary studies in SSA, we understand that its applicability and potential adoption would require empirical evaluation and testing. This is a limitation for reviews compared to implementation studies. However, we have discussed factors reported in the included primary studies hindering mental healthcare service delivery. These factors must be considered in a preliminary effort to adopt the proposed models.

For example, the study proposes training traditional healers to offer psychosocial support and mentions the mhGAP as a potential training tool. The mhGAP is a clinical protocol, designed to build capacity among healthcare practitioners who are already licensed by a recognized clinical governing body to practice in a given jurisdiction. Although the word "lay healthcare workers" is often used , it does not refer to unlicensed practitioners. Therefore, the authors might reconsider reviewing this section.

• Thank you. We acknowledge the role of the mhGAP and its application. We have argued its potential for its adaption and application to traditional healers in identifying and providing basic psychological support following this standardised clinical protocol.

---

## [Decision Letter · Decision Letter 2]

21 Apr 2025

Pathways to mental healthcare services across local health systems in sub-Saharan Africa: Findings from a Systematic Review

PONE-D-24-01443R2

Dear Dr. Baldeh,

We’re pleased to inform you that your manuscript has been judged scientifically suitable for publication and will be formally accepted for publication once it meets all outstanding technical requirements.

Kind regards,

Ryan G Wagner, MSc(Med), MBBCh, PhD

Academic Editor

PLOS ONE

Additional Editor Comments (optional):

In the copy-proof version of the manuscript, please ensure that the reference formatting is in the Journal's preferred style. Also, please adapt either British or American spelling throughout (i.e., 'behaviours' - Line 150 vs standardized' - Line 152).

Reviewers' comments:

Reviewer's Responses to Questions

**Comments to the Author**

Reviewer #3: All comments have been addressed

2. Is the manuscript technically sound, and do the data support the conclusions?

Reviewer #3: Yes

3. Has the statistical analysis been performed appropriately and rigorously?

Reviewer #3: N/A

4. Have the authors made all data underlying the findings in their manuscript fully available?

Reviewer #3: Yes

5. Is the manuscript presented in an intelligible fashion and written in standard English?

Reviewer #3: Yes

Reviewer #3: The authors have satisfactorily addressed all the previous comments providers by myself and other reviewers.

**Do you want your identity to be public for this peer review?** For information about this choice, including consent withdrawal, please see our Privacy Policy

Reviewer #3: **Yes: ** Mary Bitta

---

## [Editor Report · Acceptance letter]

PONE-D-24-01443R2

PLOS ONE

Dear Dr. Baldeh,

I'm pleased to inform you that your manuscript has been deemed suitable for publication in PLOS ONE. Congratulations! Your manuscript is now being handed over to our production team.

Kind regards,

on behalf of

Prof. Ryan G Wagner

Academic Editor

PLOS ONE